# Fast Computation and Optimization for Opinion-Based Quantities of Friedkin-Johnsen Model

**Haoxin Sun, Yubo Sun, Xiaotian Zhou, Zhongzhi Zhang**[*]
College of Computer Science and Artificial Intelligence
Fudan University
{23110240089,25110890019}@m.fudan.edu.cn, {20210240043,zhangzz}@fudan.edu.cn

## Abstract

In this paper, we address the problem of fast computation and optimization of opinion-based quantities in the Friedkin–Johnsen (FJ) model. We first introduce the concept of partial rooted forests, based on which we present an efficient algorithm for computing relevant quantities using this method. Furthermore, we study two optimization problems in the FJ model: the Opinion Minimization Problem and the Polarization and Disagreement Minimization Problem. For both problems, we propose fast algorithms based on partial rooted forest samplings. Our methods reduce the time complexity from linear to sublinear. Extensive experiments on real-world networks demonstrate that our algorithms are both accurate and efficient, outperforming state-of-the-art methods and scaling effectively to large-scale networks.

## 1 Introduction

Online social networks and social media have become integral to our daily lives, fundamentally altering how individuals communicate, exchange, and form opinions [33, 21, 48, 50, 59, 44]. Recent studies suggest that, unlike traditional forms of communication, online interactions in the digital era have profoundly impacted human behavior, facilitating the widespread, critical, and complex propagation of information [40]. To better understand the mechanisms driving opinion formation and dissemination, various mathematical frameworks for opinion dynamics have been developed [27, 41, 17, 6]. Among these models, the Friedkin-Johnsen (FJ) model [19] stands out as one of the most widely used, with applications spanning multiple fields [8, 20]. For instance, a modified FJ model was recently used to study the Paris Agreement negotiations, revealing key factors behind the achieved consensus [8].

The fundamental concept in opinion dynamics is the opinion itself, which serves as the basis for many opinion-based quantities that have garnered significant attention. Among these quantities are the overall opinion, which reflects public sentiment on specific issues, and various social phenomena such as polarization, disagreement, and conflict. Given the importance of these opinion-based quantities, a key challenge is how to effectively compute and optimize them. A Laplacian solver-based approach for computing these quantities was proposed in [52, 54], followed by a sampling-based algorithm to accelerate the computation process [39]. For the optimization of opinion-based quantities, a range of methods has been introduced, including matrix inversion [24], vector projection [56, 59], eigencentralities [5], convex optimization [38], and sampling techniques [44]. Although many effective methods have been proposed for computing and optimizing opinion-based quantities in the FJ model, they are often limited to specific problems or suffer from high time complexity. Consequently, a unified framework to efficiently address both computation and optimization of opinion-based quantities in the FJ model is imperative.

---

[*]Corresponding author.

39th Conference on Neural Information Processing Systems (NeurIPS 2025).

In this paper, we propose several algorithms to compute and optimize the opinion-based quantities. Our main contributions are as follows:

- We introduce the concept of partial rooted forests and, based on which we propose a fast sampling-based algorithm for computing opinion-based quantities. Our methods effectively capture the essential structural information of the graph, ensuring both efficiency and accuracy.

- We address two optimization problems in the FJ model: minimizing a weighted average of expressed opinions, and reducing polarization and disagreement via edge addition. For both problems, we design fast algorithms based on partial rooted forest samplings, reducing the time complexity from linear to sublinear.

- We conduct extensive experiments on a variety of real-world networks, which shows that our algorithms demonstrate both high accuracy and efficiency compared to state-of-the-art methods and are scalable to large networks.

## 2 Related Work

Mathematical modeling plays a crucial role in understanding opinion dynamics, and numerous models have been proposed over the years to capture various aspects of opinion formation [27, 41, 17, 6]. Among these, the Friedkin-Johnsen (FJ) model [19] stands out as a foundational model, building upon and extending the DeGroot model [16]. Given its theoretical importance and practical applications, the FJ model has garnered significant attention since its introduction. A sufficient stability condition for the FJ model was derived in [42], and the model's average innate opinion was characterized in [15]. Additionally, the vector of expressed opinions at equilibrium was formulated in [15, 10]. Further interpretations and insights into the FJ model have also been provided [23, 10].

The sum of opinions has attracted significant attention, with various research groups addressing the optimization problem of maximizing overall opinion through leader selection [55, 26, 57] or link recommendation [56, 58] based on the DeGroot model. In the case of the FJ model, node-based strategies have been proposed over the past decade to optimize the sum of opinions on unsigned graphs, including modifications to initial opinions [3], the expression of opinions [24, 44], and sensitivity to persuasion [2, 11, 1, 36]. Our solution for optimizing the average expressed opinion in the FJ model is both more efficient and effective compared to the state-of-the-art approach presented in [44].

The explosive growth of social media and online social networks has given rise to several social phenomena, including polarization [37, 38, 4], disagreement [38], filter bubbles [7, 30], conflicts [14], and controversies [14]. In response to these challenges, research has evolved in various directions, with fast algorithms being developed to efficiently compute these quantities [52, 54, 39]. Some studies have focused on identifying groups of users with an open attitude toward opposing information [22, 18], aiming to connect users with differing opinions in order to mitigate the filter bubble effect [47, 59, 5, 38, 60]. In this paper, we study the problem of computing opinion-based quantities in the FJ model [39, 52] and two optimization problems in the FJ model [44, 59]. Our proposed algorithms demonstrated greater efficiency and effectiveness compared to the state-of-the-art methods in [39, 44, 59].

## 3 Preliminaries

### 3.1 Graph and Laplacian Matrix

Let $\mathcal{G} = (V, E)$ denote an unweighted simple undirected graph, which consists of $n = |V|$ nodes and $m = |E|$ edges. An edge $(v_i, v_j) \in E$ indicates the edge between node $v_i$ and node $v_j$. In what follows, $v_i$ and $i$ are used interchangeably to represent node $v_i$ if incurring no confusion. The structure of graph $\mathcal{G} = (V, E)$ is captured by its adjacency matrix $\boldsymbol{A} = (a_{ij})_{n \times n}$, where $a_{ij} = a_{ji} = 1$ if there is an edge between node $v_i$ and node $v_j$ and $0$ otherwise. The degree $d_i$ of node $i$ is defined by $d_i = \sum_{j=1}^{n} a_{ij}$. The diagonal degree matrix representing the degrees of graph $\mathcal{G}$ is $\boldsymbol{D} = \text{diag}(d_1, d_2, \ldots, d_n)$, and the Laplacian matrix is $\boldsymbol{L} = \boldsymbol{D} - \boldsymbol{A}$. For any given node $i$, $N_i$ denotes the set of its neighbors, meaning $N_i = \{j : (i, j) \in E\}$. A path $P$ from node $v_1$ to $v_j$ is a sequence of alternating nodes and edges $v_1, (v_1, v_2), v_2, \cdots, v_{j-1}, (v_{j-1}, v_j), v_j$ where each node is

unique and every edges connects $v_i$ to $v_{i+1}$. A loop is a path plus an edge from the ending node to the starting node.

## 3.2 Friedkin-Johnsen Model

The Friedkin-Johnsen (FJ) model [19] is a widely used framework for modeling opinion evolution and formation. In the FJ model applied to a graph $\mathcal{G} = (V, E)$, each node (or agent) $i \in V$ is characterized by two types of opinions: an internal opinion $s_i$ and an expressed opinion $z_i(t)$ at time $t$. The internal opinion $s_i \in [0, 1]$ reflects the inherent stance of node $i$ on a particular topic. Throughout the opinion evolution process, the internal opinion $s_i$ remains fixed, while the expressed opinion $z_i(t)$ evolves at time $t + 1$ as $z_i(t+1) = (s_i + \sum_{j \in N_i} a_{ij} z_j(t))/(1 + \sum_{j \in N_i} a_{ij})$.

Let $\boldsymbol{s} = (s_1, s_2, \cdots, s_n)^\top$ denote the vector of internal opinions, and let $\boldsymbol{z}(t) = (z_1(t), z_2(t), \cdots, z_n(t))^\top$ denote the vector of expressed opinions at time $t$. According to [10], as $t$ approaches infinity, $\boldsymbol{z}(t)$ converges to an equilibrium vector $\boldsymbol{z} = (z_1, z_2, \cdots, z_n)^\top$ satisfying $\boldsymbol{z} = (\boldsymbol{I} + \boldsymbol{L})^{-1}\boldsymbol{s}$. Define $\boldsymbol{\Omega} = (\boldsymbol{I} + \boldsymbol{L})^{-1} = (\omega_{ij})_{n \times n}$, referred to as the forest matrix [12, 13]. The forest matrix $\boldsymbol{\Omega}$ is doubly stochastic for undirected graphs, with all its components in the interval $[0, 1]$. Furthermore, for each column, the diagonal elements are greater than the off-diagonal elements, that is $0 \leq \omega_{ji} < \omega_{ii} \leq 1$ for any pair of different nodes $i$ and $j$, and the diagonal element $\omega_{ii}$ of matrix $\boldsymbol{\Omega}$ satisfies $\frac{1}{1+d_i} \leq \omega_{ii} \leq \frac{2}{2+d_i}$ [44]. The forest matrix $\boldsymbol{\Omega}$ serves as the fundamental matrix of the FJ model for opinion dynamics [24]. For every node $i \in V$, its expressed opinion $z_i$ is given by $z_i = \sum_{j=1}^{n} \omega_{ij} s_j$, a convex combination of the internal opinions for all nodes.

# 4 Partial Rooted Forest Samplings for Estimating the Forest Matrix

The forest matrix $\boldsymbol{\Omega}$ is the fundamental matrix of the FJ model and plays a key role in its computation and optimization. Existing methods [45, 46, 44] rely on generating rooted spanning forests over the entire graph, requiring $O(ln)$ time. We propose a local approach, called partial rooted forest samplings, as a natural extension of previous methods by using absorbing random walks. It avoids full-graph traversal and offers improved efficiency and scalability.

## 4.1 Absorbing Random Walk

The forest matrix $\boldsymbol{\Omega}$ is doubly stochastic for undirected graphs. We initiate a random walk from node $i$ and assume the walk is currently at node $k \in V$. At node $k$, the walk has a probability of $\frac{1}{1+d_k}$ to stop, and with probability $1 - \frac{1}{1+d_k}$, it moves to a randomly selected neighbor of node $k$. If the walk eventually stops at node $q$, we say that the walk has been absorbed by $q$. In this context, $\omega_{ij}$ represents the probability that a walk starting at node $i$ will eventually be absorbed at node $j$.

To formally explain this, we expand the forest matrix as an infinite series. Define $\boldsymbol{P} = (\boldsymbol{I} + \boldsymbol{D})^{-1}\boldsymbol{A}$, then the forest matrix has the following form: $\boldsymbol{\Omega} = (\boldsymbol{I} - (\boldsymbol{I} + \boldsymbol{D})^{-1}\boldsymbol{A})^{-1}(\boldsymbol{I} + \boldsymbol{D})^{-1} = (\boldsymbol{I} - \boldsymbol{P})^{-1}(\boldsymbol{I} + \boldsymbol{D})^{-1} = \sum_{k \geq 0} \boldsymbol{P}^k (\boldsymbol{I} + \boldsymbol{D})^{-1}$. Here, the $i, j$-th entry of $\boldsymbol{P}^k$ represents the probability that a walk starting at $i$ takes exactly $k$ steps to reach $j$. Multiplying this by $\frac{1}{1+d_j}$ gives the probability that the walk takes $k$ steps and is then absorbed at $j$. Then $\omega_{ij}$ can be expressed as $\omega_{ij} = \boldsymbol{e}_i^\top \boldsymbol{\Omega} \boldsymbol{e}_j = \sum_{k \geq 0} \boldsymbol{e}_i^\top \boldsymbol{P}^k (\boldsymbol{I} + \boldsymbol{D})^{-1} \boldsymbol{e}_j = \frac{1}{1+d_j} \sum_{k \geq 0} \boldsymbol{e}_i^\top \boldsymbol{P}^k \boldsymbol{e}_j$. Choose an initial node $i \in V$, and perform the absorbing random walk several times. Using the probabilistic interpretation of the forest matrix, where $\omega_{ij}$ represents the probability of a random walk starting at node $i$ and being absorbed at node $j$, we can estimate the $i$-th row of the forest matrix. The following lemma establishes the expected time complexity of the absorbing random walk:

**Lemma 4.1** *For an undirected graph $\mathcal{G} = (V, E)$ and its related forest matrix $\boldsymbol{\Omega}$, the expected length of an absorbing random walk starting at node $i$ is $\sum_{j=1}^{n} \omega_{ij} d_j$. If the initial node $i$ is chosen randomly, the expected length of the random walk to absorption is $\bar{d} = \frac{1}{n} \sum_{i=1}^{n} d_i = \frac{2m}{n}$.*

**Proof.** Let $l_i$ denote the expected length of an absorbing random walk starting at node $i$, and define the vector $\boldsymbol{l} = (l_1, \cdots, l_n)^\top$. For any $i \in V$, the relationship between $l_i$ and its neighbors is given by $l_i = \frac{d_i}{1+d_i}(1 + \frac{1}{d_i}\sum_{j \in N_i^+} l_j)$. Rewriting this in matrix form, we have $\boldsymbol{l} = (\boldsymbol{I} + \boldsymbol{D})^{-1}\boldsymbol{D}\boldsymbol{1} +$

$(\boldsymbol{I} + \boldsymbol{D})^{-1}\boldsymbol{Al}$. Solving this equation yields $\boldsymbol{l} = \boldsymbol{\Omega D1}$. Thus, for each $i \in V$, the expected length is $l_i = \sum_{j=1}^{n} \omega_{ij}d_j$. If the initial node $i$ is chosen uniformly at random, the expected length of the random walk is $\bar{l} = \frac{1}{n}\sum_{i=1}^{n} l_i = \frac{1}{n}\sum_{i=1}^{n}\sum_{j=1}^{n} \omega_{ij}d_j = \frac{1}{n}\sum_{j=1}^{n}(\sum_{i=1}^{n} \omega_{ij})d_j = \bar{d}$, where the property $\sum_{i=1}^{n} \omega_{ij} = 1$ in undirected graphs is used. $\square$

By Lemma 4.1, we can estimate $\boldsymbol{z}$ by performing absorbing random walks from each node in $V$ multiple times. However, this method requires iterating over all nodes, which is computationally inefficient.

## 4.2 Partial Rooted Forest Sampling

In this subsection, we introduce a sampling method for constructing a partial rooted forest, which significantly reduces sampling time. We first define the concept of a partial rooted forest. In an undirected graph $\mathcal{G} = (V, E)$, a rooted tree of $\mathcal{G}$ is a connected subgraph without cycle, where one node is set to be the root. An isolated node is considered as a tree with the root being itself. A partial rooted forest $\phi = (V_\phi, E_\phi)$ is a subgraph of $G$, where all connected components are rooted trees. The root set $\mathcal{R}(\phi)$ of $\phi$ is defined as the collection of root nodes from all rooted trees in $\phi$. Since each node $i$ in $V_\phi$ belongs to a specific rooted tree, we define a function $r_\phi(i) : V_\phi \to \mathcal{R}(\phi)$ mapping node $i$ to the root of its corresponding rooted tree. If $r_\phi(i) = j$, then $j \in \mathcal{R}(\phi)$, and both $i$ and $j$ belong to the same rooted tree in $\phi$.

Next, we describe the procedure for generating a partial rooted forest using a loop-erased absorbing random walk. This involves the loop-erasure technique, which eliminates loops from a random walk in chronological order [32, 31]. By applying this technique, we can utilize the paths generated by the absorbing random walk instead of discarding them. The procedure is as follows:

(i) **Initialization:** Let $\boldsymbol{S} = \{v_1, \ldots, v_p\} \subset V$ be a set of $p \geq 2$ nodes. Initialize $\phi = (V_\phi, E_\phi) = (\emptyset, \emptyset)$ and set the indicator $i = 1$.

(ii) **Performing absorbing Random Walk:** Select the first node $u = v_i$. Perform an absorbing random walk starting from $u$. Suppose that the walk is currently at a node $k \notin V_\phi$, and the walk is absorbed with probability $\frac{1}{1+d_k}$. If the node is absorbed at $k$, $k$ is marked as a root node. Otherwise, with probability $1 - \frac{1}{1+d_k}$, the walk moves to a uniformly chosen neighbor of $k$. If the walk reaches a node $k \in V_\phi$, it is immediately absorbed.

(iii) **Loop Erasure:** Let $P_u$ represent the trajectory of the walk from node $u$ to its absorption point. Apply the loop-erasure technique to $P_u$ to obtain a simple path $\hat{P}_u$. Add the nodes and edges from $\hat{P}_u$ to $V_\phi$ and $E_\phi$, respectively.

(iv) **Iteration:** If $i < p$, increment $i$ by 1 and repeat step (ii). Otherwise, terminate the process and return the partial rooted forest $\phi$.

The procedure is detailed in Algorithm PFS (Partial Rooted Forest Sampling); due to space constraints, the pseudocode is provided in the appendix.

The following theorem establishes the expected time complexity of the Partial Rooted Forest Sampling (PFS) algorithm. The time complexity is $O(rpl)$, where $r$ is a ratio dependent on the graph structure, with an upper bound $\bar{d}$. In real-world web and social networks, the average degree is typically $O(\log n)$ or a constant [49, 53]. Furthermore, our experimental results demonstrate that $r$ is consistently smaller than $\bar{d}$. These observations collectively indicate the efficiency of our algorithm in practical scenarios.

**Theorem 4.2** *For an undirected graph $\mathcal{G} = (V, E)$ and a set of $p$ nodes $\boldsymbol{S} \subset V$, the expected time complexity of Algorithm PFS is $O(rpl)$, where $r$ is the ratio of the expected number of nodes in a partial rooted forest $\phi \in L$ to $p$, and $l$ is the number of samples. $r$ satisfies relation $1 \leq r \leq \bar{d}$ if we randomly sample the set $\boldsymbol{S}$ from $V$. Algorithm PFS is more efficient than performing absorbing random walks individually for each node in $\boldsymbol{S}$.*

**Proof.** The algorithm repeats the sampling process $l$ times and outputs a list of partial rooted forests. To establish the expected time complexity, consider a single partial rooted forest $\phi$ in the list $L$. Let $q = rp$ represent the expected number of nodes in $\phi$. We will show that the expected time complexity for generating $\phi$ is $O(q)$.

First, we extend the partial rooted forest $\phi$ to a complete spanning rooted forest $\phi'$ by executing the procedure described in lines 4–16 of Algorithm 4 for the remaining nodes in $V \setminus V_\phi$. According to [44], the expected time complexity of sampling the complete rooted forest $\phi'$ is $O(n)$. Furthermore, Wilson [51] demonstrates that the order of node processing does not affect the generation of a spanning rooted forest. Hence, the partial rooted forest $\phi$ can be viewed as a subgraph of one complete rooted forest $\phi'$.

Next, we analyze the expected time complexity of generating the partial rooted forest $\phi$. Marchel [35] shows that the expected absorption time for each node in a rooted forest is $\omega_{ii}(1 + d_i)$. Sun [44] refines this result, proving that $\omega_{ii}(1 + d_i) \leq \frac{2(1+d_i)}{2+d_i} \leq 2$, which is $O(1)$.

Therefore, the expected time complexity for sampling a single partial rooted forest $\phi$ is $O(q)$, leading to a total expected time complexity of $O(ql)$ for the algorithm.

To give a rough estimation of $r$, if we sample $S$ randomly from $V$, an upper bound of $r$ is $O(\bar{d})$ according to Lemma 4.1. Moreover, Algorithm 4 retains only the necessary branches in the forest for each node. The sampling process terminates as soon as the walk reaches a node already in $V_\phi$, making it more efficient than performing absorbing random walks for all nodes in $S$ individually, as described in Subsection 4.1. $\quad\square$

## 5 Fast Sampling Method for Opinion-Based Quantities in FJ Model

### 5.1 Definitions of Opinion-Based Quantities

The FJ model includes several important quantities for analyzing the properties of social groups. Below, we provide a concise overview of these measures, following prior works [38, 37, 52].

Consider an undirected graph $\mathcal{G} = (V, E)$. Disagreement $D(\mathcal{G})$ is a measure of the difference in opinions between neighbors, calculated as $D(\mathcal{G}) = \sum_{(i,j)\in E}(z_i - z_j)^2 = \boldsymbol{z}^\top \boldsymbol{L} \boldsymbol{z}$. Polarization $P(\mathcal{G})$ reflects how far individual opinions deviate from the average-expressed opinion. Using the vector $\bar{\boldsymbol{z}} = \boldsymbol{z} - \frac{\boldsymbol{z}^\top \mathbf{1}}{n}\mathbf{1}$, polarization is defined as $P(\mathcal{G}) = \sum_{i\in V}(z_i - \bar{z})^2 = \bar{\boldsymbol{z}}^\top \bar{\boldsymbol{z}}$. Internal conflict $I(\mathcal{G})$ quantifies the discrepancy between the expressed opinions $\boldsymbol{z}$ and the initial opinions $\boldsymbol{s}$, given by $I(\mathcal{G}) = \sum_{i\in V}(z_i - s_i)^2 = \boldsymbol{z}^\top \boldsymbol{L}^2 \boldsymbol{z}$. Controversy $C(\mathcal{G})$ represents the overall magnitude of expressed opinions $C(\mathcal{G}) = \sum_{i\in V} z_i^2 = \boldsymbol{z}^\top \boldsymbol{z}$. Disagreement-controversy $DC(\mathcal{G})$ is a combined measure capturing both disagreement and controversy $DC(\mathcal{G}) = D(\mathcal{G}) + C(\mathcal{G}) = \sum_{i\in V} s_i z_i = \boldsymbol{s}^\top \boldsymbol{z}$.

### 5.2 Estimating Opinion-Based Quantities

In this subsection, we present a sampling-based algorithm for QE (Opinion-Based Quantities Estimation). The algorithm leverages the partial rooted forest samplings method to estimate the opinion-based quantities efficiently. We first provide a lemma to show that the map between the nodes in $S$ to its root nodes in the partial rooted forest can be used to estimate the opinion-based quantities.

**Lemma 5.1** *For an undirected graph $\mathcal{G} = (V, E)$, a set of nodes $S \subset V$, and internal opinion vectors $\boldsymbol{s} = (s_1, \cdots, s_n)^\top$, let $\phi$ be the partial rooted forest generated by Algorithm PFS . For node $i \in S$, let $r_\phi(i)$ denote the root node of $i$ in $\phi$. Then, the estimator $\widehat{z}_i = \sum_{j=1}^n \mathbb{I}_{\{r_\phi(i)=j\}} s_j$ is an unbiased estimator of $z_i$.*

Lemma 5.1 shows that the estimator $\widehat{z}_i$ is unbiased for $z_i$. By applying the partial rooted forest samplings method, we can estimate the opinion-based quantities in the FJ opinion dynamics model, and we now give the pseudocode for the sampling-based algorithm.

Algorithm 1 PF-QE (Partial Rooted Forest Method for QE ) provides the pseudocode for estimating the opinion-based quantities in the FJ opinion dynamics model. The algorithm leverages the partial rooted forest samplings method to estimate the opinion-based quantities efficiently. Instead of estimating all nodes in $V$, algorithm 1 focuses on a set of nodes $S$ with $p$ nodes, and samples $l$ partial rooted forests. Now we provide a theorem, showing the parameters $p$ and $l$ for a given error tolerance.

**Theorem 5.2** *For an undirected graph $\mathcal{G} = (V, E)$ and internal opinion vectors $\boldsymbol{s} = (s_1, \cdots, s_n)^\top$, if we choose $p = O(\frac{1}{\epsilon^2}\log\frac{1}{\delta})$, $l = O(\frac{1}{\epsilon^2}\log\frac{1}{\delta})$, and node set $S$ is randomly sampled from $V$, then*

**Algorithm 1:** PF-QE($\mathcal{G}, \boldsymbol{S}, \boldsymbol{s}, L$)

---

**Input** : Graph $\mathcal{G} = (V, E)$, node set $\boldsymbol{S} = \{v_1, \ldots, v_p\}$, internal opinion $\boldsymbol{s} = (s_1, \cdots, s_n)^\top$
**Output** : Estimates of opinion-based quantities

1   Sample a list $L$ of $l$ partial rooted forests
2   Initialize $\widehat{\boldsymbol{z}}_{\boldsymbol{S}} \leftarrow \boldsymbol{0}$ (a $p$-dimensional vector for storing $\widehat{z}_i$ for each $v_i \in \boldsymbol{S}$)
3   **for** $k \leftarrow 1$ **to** $l$ **do**
4      $\phi \leftarrow L[k]$
5      **for** $i \leftarrow 1$ **to** $p$ **do**
6         $r_i \leftarrow r_\phi(v_i)$, $\widehat{\boldsymbol{z}}_{\boldsymbol{S}}[i] \leftarrow \widehat{\boldsymbol{z}}_{\boldsymbol{S}}[i] + s_{r_i}$

7   $\widehat{\boldsymbol{z}}_{\boldsymbol{S}} \leftarrow \frac{\widehat{\boldsymbol{z}}_{\boldsymbol{S}}}{l}$, $\widehat{C} \leftarrow \frac{n}{p} \sum_{i=1}^{p} \widehat{\boldsymbol{z}}_{\boldsymbol{S}}[i]^2$, $\widehat{DC} \leftarrow \frac{n}{p} \sum_{i=1}^{p} s_i \widehat{\boldsymbol{z}}_{\boldsymbol{S}}[i]$, $\widehat{D} \leftarrow \widehat{DC} - \widehat{C}$
8   $\widehat{\bar{z}}_{\boldsymbol{S}} = \frac{1}{p} \sum_{i=1}^{p} \widehat{\boldsymbol{z}}_{\boldsymbol{S}}[i]$, $\widehat{P} \leftarrow \frac{n}{p} \sum_{i=1}^{p} (\widehat{\boldsymbol{z}}_{\boldsymbol{S}}[i] - \widehat{\bar{z}}_{\boldsymbol{S}})^2$, $\widehat{I} \leftarrow \frac{n}{p} \sum_{i=1}^{p} (\widehat{\boldsymbol{z}}_{\boldsymbol{S}}[i] - s_i)^2$
9   **return** $\widehat{D}, \widehat{P}, \widehat{I}, \widehat{C}, \widehat{DC}$

---

*Algorithm 1 estimates the opinion-based quantities $D(\mathcal{G})$, $P(\mathcal{G})$, $I(\mathcal{G})$, $C(\mathcal{G})$, and $DC(\mathcal{G})$ within an absolute error of $n\epsilon$ with probability at least $1 - \delta$. The total time complexity of Algorithm 1 is $O(rpl)$.*

## 6   Partial Forest Sampling Techniques for Optimization Problems in FJ model

### 6.1   Opinion Minimization Problem

> **Problem 1** *[Opinion Minimization Problem (OpMin)] Given an undirected graph $\mathcal{G} = (V, E)$, a parameter vector $\boldsymbol{c} = (c_1, \cdots, c_n)^\top$, where $c_i \in [0, 1]$, an integer $k \ll n$, and an iternal opinion vecter $\boldsymbol{s}$, we aim to find the set $H \subseteq V$ of $k$ nodes, and set their internal opinions to 0, so that the function $f(\boldsymbol{c}, H) = \frac{1}{n} \boldsymbol{c}^\top \boldsymbol{z} = \frac{1}{n} \sum_{i=1}^{n} c_i z_i$ is minimized. That is,*
>
> $$H = \underset{U \subseteq V, |U| = k}{\arg\min} \; f(\boldsymbol{c}, U). \tag{1}$$

Our formulation of Problem 1 generalizes the setting in [44]. Specifically, when $c_i = 1$ for all $i \in V$, it reduces to the average-expressed opinion minimization problem studied in [44]. To address this problem efficiently, we propose a fast algorithm based on partial rooted forest samplings, which reduces the time complexity from $O(ln)$ to $O(rpl)$.

Recall that $\boldsymbol{z} = (\boldsymbol{I} + \boldsymbol{L})^{-1} \boldsymbol{s}$ is the equilibrium vector of the expressed opinions. Then $z_i = \sum_{j=1}^{n} \omega_{ij} s_j$. We can rewrite the objective function as $f(\boldsymbol{c}, H) = \frac{1}{n} \sum_{i=1}^{n} c_i z_i = \frac{1}{n} \sum_{i=1}^{n} c_i \sum_{j=1}^{n} \omega_{ij} s_j = \frac{1}{n} \sum_{j=1}^{n} (\sum_{i=1}^{n} c_i \omega_{ij}) s_j$. Define $\gamma_j = \frac{1}{n} \sum_{i=1}^{n} c_i \omega_{ij}$, then the objective function can be rewritten as $f(\boldsymbol{c}, H) = \sum_{j=1}^{n} \gamma_j s_j$. In this case, the objective function is linear with respect to the internal opinions $s_i$. To solve OpMin, we need to find the set $H$ of $k$ nodes with the smallest $\gamma_j s_j$ values.

In [44], the authors use the forest sampling method to solve the problem. However, the algorithm needs to perform a random walk process across all nodes in the graph, which is computationally expensive. To overcome this, we propose an efficient method using the partial rooted forest samplings. The key to solving OpMin lies in estimating the $\gamma_j = \frac{1}{n} \sum_{i=1}^{n} c_i \omega_{ij}$ for all nodes in the graph. Suppose that we choose a set of $p$ nodes $\boldsymbol{S}$ randomly from $V$, and sample a partial rooted forest $\phi$. Then, define the estimator $\widehat{\gamma}_j(\phi) = \frac{1}{p} \sum_{i \in \boldsymbol{S}} c_i \mathbb{I}_{\{r_\phi(i) = j\}}$. We sample $l$ partial rooted forests $\phi_1, \cdots, \phi_l$, and estimate the $\gamma_j$ values using $\widehat{\gamma}_j = \frac{1}{l} \sum_{k=1}^{l} \widehat{\gamma}_j(\phi_k)$. We detail the algorithm for solving OpMin in Algorithm 2.

Algorithm 2 PF-OPMIN(Partial Rooted Forest Method for OpMin) provides the pseudocode for solving OpMin. The algorithm leverages the partial rooted forest samplings method to estimate the $\gamma_j$ values efficiently. By sampling $l$ partial rooted forests, we estimate the $\gamma_j$ values using the estimator $\widehat{\gamma}_j$. The algorithm then selects the $k$ nodes with the smallest $\widehat{\gamma}_j s_j$ values as the optimal set $\widehat{H}$.

**Algorithm 2:** PF-OPMIN($\mathcal{G}, \boldsymbol{S}, \boldsymbol{c}, l, k$)

---

**Input** :Graph $\mathcal{G} = (V, E)$, node set $\boldsymbol{S} = \{v_1, \ldots, v_p\}$, parameter vector $\boldsymbol{c} = (c_1, \cdots, c_n)^\top$,
number of forests $l$, number of nodes $k$

**Output** :Optimal set $\widehat{H}$

1 Initialize $\widehat{\gamma} \leftarrow \mathbf{0}$, $\widehat{H} \leftarrow \emptyset$. Sample a list $L$ of $l$ partial rooted forests

2 **for** $t \leftarrow 1$ **to** $l$ **do**

3      $\phi \leftarrow L[t]$

4      **for** $i \leftarrow 1$ **to** $p$ **do**

5          $j \leftarrow r_\phi(v_i), \quad \widehat{\gamma}[j] \leftarrow \widehat{\gamma}[j] + c_{v_i}/lp$

6 **for** $i \leftarrow 1$ **to** $k$ **do**

7      $j \leftarrow \arg\max_{v \in V \setminus H} \widehat{\gamma}[v] s_v, \quad \widehat{H} \leftarrow \widehat{H} \cup \{j\}$

8 **return** $\widehat{H}$

---

Consider an undirected graph $\mathcal{G} = (V, E)$ and a parameter vector $\boldsymbol{c} = (c_1, \cdots, c_n)^\top$, where each $c_i$ is a constant. By setting the parameters $p = O\left(\frac{1}{\epsilon^2} \log \frac{1}{\delta}\right)$ and $l = O\left(\frac{1}{\epsilon^2} \log \frac{1}{\delta}\right)$, and selecting the node set $\boldsymbol{S}$ through random sampling from $V$, Algorithm 2 can solve OpMin within an absolute error of $k\epsilon$, with probability at least $1 - \delta$. Specifically, the approximation satisfies $f(\boldsymbol{c}, H) - f(\boldsymbol{c}, \widehat{H}) \leq k\epsilon$. This result follows a proof strategy analogous to that of Theorem 5.2 and Theorem 5.5 in [44], where $r$ is the ratio of the expected number of nodes in a partial rooted forest $\phi \in L$ to $p$. Our algorithm runs in $O(rpl)$ time and outperforms the state-of-the-art method [44], which requires $O(ln)$ time. Since $p \ll n$ and, as shown in the experimental section, the empirical results on real-world networks indicate that $r < 15$, our approach improves the time complexity from linear to sublinear.

## 6.2 Polarization and Disagreement Minimization problem

In this subsection, we consider the Polarization and Disagreement Minimization Problem proposed in [59]. The problem focuses on selecting a set of edges not present in the original graph to minimize the sum of polarization and disagreement. We define the P-D index as $\mathcal{I}(\mathcal{G}) = D(\mathcal{G}) + P(\mathcal{G}) = \bar{\boldsymbol{s}}^\top \boldsymbol{\Omega} \bar{\boldsymbol{s}}$, and denote the augmented graph as $\mathcal{G} + T = (V, E \cup T)$. The problem is formally described as follows:

> **Problem 2** *[Polarization and Disagreement Minimization Problem (PDMin)] Given an undirected graph $\mathcal{G} = (V, E)$ and a candidate edge set $E_C \subseteq \binom{V}{2} \setminus E$, the goal is to select a subset $T \subseteq E_C$ of $k \ll n$ edges to add to the graph, such that the P-D index of the new graph is minimized. Define the objective function as $f(T) = \mathcal{I}(\mathcal{G}) - \mathcal{I}(\mathcal{G} + T)$, and find*
> $$T = \arg\max_{T \subseteq E_C, |T| = k} f(T). \tag{2}$$

In [59], the authors show that the problem is combinatorial in nature and propose a greedy algorithm to solve it. They prove that for a candidate edge $e \in E_C$ connecting nodes $u$ and $v$, with vector $\boldsymbol{b}_e = \boldsymbol{e}_u - \boldsymbol{e}_v$, the marginal gain is given by $f(e) = \frac{\boldsymbol{s}^\top \boldsymbol{\Omega} \boldsymbol{b}_e \boldsymbol{b}_e^\top \boldsymbol{\Omega} \boldsymbol{s}}{1 + \boldsymbol{b}_e^\top \boldsymbol{\Omega} \boldsymbol{b}_e} = \frac{(z_u - z_v)^2}{1 + r_{uv}} \geq 0$, where $r_{uv}$ is the forest distance between nodes $u$ and $v$, defined as $r_{uv} \triangleq \boldsymbol{b}_{uv}^\top \boldsymbol{\Omega} \boldsymbol{b}_{uv}$. A greedy algorithm computes the marginal gain for each candidate edge in $E_C$, selects the one with the largest gain, and repeats this process $k$ times to obtain the final set $T$.

We set the sample node set for partial rooted forests to be the set of all nodes that appear in the candidate edge set $E_C$, that is, $\boldsymbol{S} = \bigcup_{(i,j) \in E_C} \{i, j\}$. Then, we generate $l$ partial rooted forests $\{\phi_1, \cdots, \phi_l\}$ and use them to estimate the marginal gain $f(e)$ for each candidate edge $e \in E_C$. The detailed procedure is presented in Algorithm PF-PDMIN(Partial Rooted Forest Method for PDMin). For any candidate edge $e$, if we set $l = O\left(\frac{1}{\epsilon^2} \log \frac{1}{\delta}\right)$, $|\widehat{f}(e) - f(e)| < \epsilon$ is guaranteed to hold with probability at least $1 - \delta$, by applying Hoeffding's inequality [25]. In [59], the authors use the Johnson–Lindenstrauss Lemma [28] and a fast Laplacian solver [43] to approximate the marginal

**Algorithm 3:** PF-PDMIN($\mathcal{G}, E_C, \boldsymbol{s}, l, k$)

**Input** : Graph $\mathcal{G} = (V, E)$, candidate edge set $E_C$, internal opinion vector $\boldsymbol{s} = (s_1, \cdots, s_n)^\top$, number of forests $l$, number of edges $k$

**Output** : Selected edge set $\widehat{T}$

1 Initialize $\widehat{T} \leftarrow \emptyset$, $\boldsymbol{S} \leftarrow \bigcup_{(i,j) \in E_C} \{i, j\}$

2 **for** $t \leftarrow 1$ **to** $k$ **do**

3      Sample $l$ partial rooted forests $\{\phi_1, \ldots, \phi_l\}$ on node set $\boldsymbol{S}$

4      **foreach** $(u, v) \in E_C \setminus \widehat{T}$ **do**

5          Initialize $\widehat{z}_u \leftarrow 0, \widehat{z}_v \leftarrow 0, \widehat{r}_{uv} \leftarrow 0$

6          **for** $i \leftarrow 1$ **to** $l$ **do**

7              $r_u \leftarrow r_{\phi_i}(u), \quad r_v \leftarrow r_{\phi_i}(v), \quad \widehat{z}_u \leftarrow \widehat{z}_u + s_{r_u}, \quad \widehat{z}_v \leftarrow \widehat{z}_v + s_{r_v}$

8              $\widehat{r}_{uv} \leftarrow \widehat{r}_{uv} + \mathbb{I}_{\{u=r_u\}} + \mathbb{I}_{\{v=r_v\}} - \mathbb{I}_{\{u=r_v\}} - \mathbb{I}_{\{v=r_u\}}$

9          $\widehat{z}_u \leftarrow \widehat{z}_u/l, \quad \widehat{z}_v \leftarrow \widehat{z}_v/l, \quad \widehat{r}_{uv} \leftarrow \widehat{r}_{uv}/l, \quad \widehat{f}(u,v) \leftarrow (\widehat{z}_u - \widehat{z}_v)^2/(1 + \widehat{r}_{uv})$

10      $(u^*, v^*) \leftarrow \arg\max_{(u,v) \in E_C \setminus T} \widehat{f}(u,v)$

11      $\widehat{T} \leftarrow \widehat{T} \cup \{(u^*, v^*)\}, \quad E \leftarrow E \cup \{(u^*, v^*)\}, \quad V \leftarrow V \cup \{u^*, v^*\}$

12 **return** $T$

gain, resulting in an overall time complexity of $\widetilde{O}(mk)$. In contrast, we leverage partial rooted forest samplings to reduce the time complexity to $O(r|E_C|kl)$.

## 7 Experiments

In this section, we conduct extensive experiments on various real-life networks in order to evaluate the performance of our algorithms, in terms of accuracy and efficiency. Our source code is publicly available on `https://github.com/HaoxinSun98/FJ-PF`.

**Datasets and Equipment.** The datasets for chosen real networks are accessed publicly through KONECT [29] and SNAP [34]. Our experiments cover a varied selection of real-world networks, and the specifics of these datasets are outlined in Table 1. All experiments are carried out using the Julia programming language within a computational environment equipped with a 2.5 GHz Intel E5-2682v4 CPU and 256GB of primary memory.

Table 1: Statistics of the datasets used in the experiments, including the number of nodes, number of edges, average degree, and the parameter $r$, which denotes the ratio of the average number of nodes in a partial rooted forest.

| Network | Nodes | Edges | $\bar{d}$ | $r$ |
|---|---|---|---|---|
| Delicious | 536,108 | 1,365,961 | 5.1 | 2.4 |
| Youtube | 1,134,890 | 2,987,624 | 5.3 | 2.4 |
| Pokec | 1,632,803 | 22,301,964 | 27.3 | 9.0 |
| Orkut | 3,072,441 | 117,184,899 | 76.3 | 13.6 |
| Livejournal | 7,489,073 | 112,305,407 | 30.0 | 3.3 |
| Twitter | 41,652,230 | 1,202,513,046 | 57.7 | 7.7 |

**Baselines.** For the FJ quantities estimation problem, we compare our proposed algorithms with the LapSolver [52] and LazyWalk [39]. For OpMin, we compare our algorithm with the forest sampling method FAST [44]. For PDMin, we compare our proposed algorithms with the greedy algorithm FASTGREEDY [59].

### 7.1 Estimation for Relevant Quantities

In this subsection, we evaluate the performance of our proposed algorithms for estimating relevant quantities for FJ model on real-world networks. We compare our algorithms with LapSolver [52]

Table 2: Time complexity of our algorithms and baselines for three problems with parameter settings.

| QE | | OpMin | | PDmin | | Settings |
|---|---|---|---|---|---|---|
| PF-QE | $O(rpl)$ | PF-OPMIN | $O(rpl)$ | PF-PDMIN | $O(rkl\|E_C\|)$ | $r < 15\,, l = 10^3\,, p = 10^4$ |
| LazyWalk | $O(r'pl')$ | FAST | $O(ln)$ | FASTGREEDY | $\widetilde{O}(mk)$ | $r' = 4 \times 10^3, l' = 600$ |
| LapSolver | $\widetilde{O}(m)$ | EXACT | $\widetilde{O}(m)$ | — | — | $k = 50, \|E_C\| = 10^4$ |

and LazyWalk [39]. LapSolver achieves high accuracy [52] with a $10^{-6}$ relative error and is treated as the ground truth. Following the setting in [39], we set the number of samples to $p = 10,000$ and vary the sampling parameters to compare the running time and mean relative error of the expressed opinions computed by PF-QE and LazyWalk. The results are shown in Figure 1. Both PF-QE and LazyWalk are sampling-based algorithms and are parallelized using 8 threads. Each experiment is repeated 10 times, and the average results are reported.

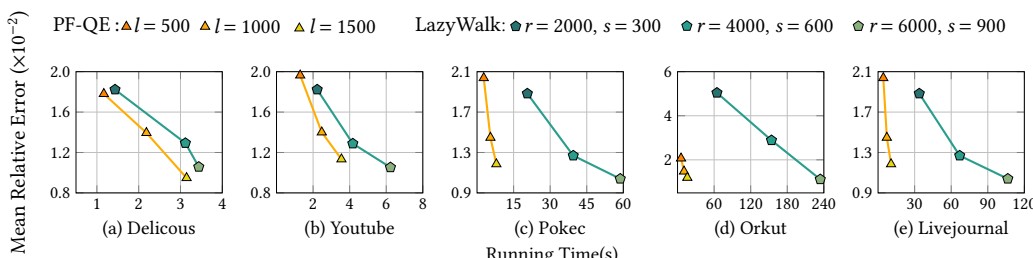

Figure 1: Comparison of mean relative error for $z_i$ and running time under varying parameters for PF-QE and LazyWalk. Internal opinions are generated using the uniform distribution.

From Figure 1, we observe that the PF-QE curves consistently lie below and to the left of those for LazyWalk, indicating that PF-QE achieves better accuracy and lower running time. Based on this observation, we fix $l = 1000$ for PF-QE, and for LazyWalk, we use 4,000 walks with 600 steps, following the configuration in [39]. For other quantities, we maintain 8-thread parallelization, repeat each experiment 10 times, and present the averaged results in Table 3.

Table 3: Mean relative errors and running time for three algorithms on six networks. Internal opinions are generated using the uniform distribution.

| Network | Time(s) | | | Reletive Error in % | | | | | | | | | |
|---|---|---|---|---|---|---|---|---|---|---|---|---|---|
| | LapSolver | LazyWalk | PF-QE | LazyWalk | | | | | PF-QE | | | | |
| | | | | $P$ | $I$ | $C$ | $DC$ | $D$ | $P$ | $I$ | $C$ | $DC$ | $D$ |
| Delicious | 7.2 | 3.1 | 2.2 | 0.90 | 0.48 | 0.83 | 0.42 | 3.58 | 1.38 | 0.42 | 0.55 | 0.32 | 4.05 |
| Youtube | 9.6 | 4.1 | 2.4 | 1.03 | 0.75 | 0.21 | 0.33 | 3.52 | 0.95 | 0.73 | 0.22 | 0.31 | 4.19 |
| Pokec | 49.8 | 39.5 | 5.3 | 1.83 | 0.93 | 0.25 | 0.50 | 11.01 | 2.01 | 0.96 | 0.21 | 0.51 | 10.63 |
| Orkut | 291.9 | 142.2 | 9.9 | 3.97 | 0.62 | 5.43 | 2.71 | 34.77 | 2.98 | 0.72 | 0.23 | 0.47 | 13.67 |
| Livejournal | 186.8 | 67.6 | 6.9 | 1.37 | 0.85 | 0.31 | 0.61 | 8.36 | 1.30 | 0.90 | 0.30 | 0.52 | 8.17 |
| Twitter | - | 160.6 | 32.3 | - | - | - | - | - | - | - | - | - | - |

The results show that our algorithms achieve relative errors below $1.5\%$ for average expressed opinion and below $4\%$ for $P$, $I$, $C$, and $DC$. Errors for $D$ are higher due to its dependence on both $DC$ and $C$, amplifying estimation errors. PF runs faster than both LapSolver and LazyWalk while maintaining competitive accuracy. For the Twitter network, LapSolver cannot process the data due to time and memory constraints, whereas our algorithm handles it efficiently, running approximately five times faster than LazyWalk.

## 7.2 Optimization Problems

In this subsection, we evaluate the performance of our proposed algorithms for solving the FJ optimization problems OpMin and PDMin on real-world networks. We set $l = 1000$, $p = 10000$,

and $k = 50$ for our algorithms. Since our algorithms are sampling-based, we repeat each experiment 10 times and report the average results to ensure robustness and consistency.

Table 4: Running time and effectiveness of OPMIN (in terms of opinion decline $\theta$) and PDMIN (in terms of P-D index decline $\delta$).

| Network | OpMin | | | | | | PDMin | | | |
|---|---|---|---|---|---|---|---|---|---|---|
| | PF-OPMIN | | FAST | | EXACT | | PF-PDMIN | | FASTGREEDY | |
| | time | $\theta$ | time | $\theta$ | time | $\theta$ | time | $\delta$ | time | $\delta$ |
| Delicious | 2.2 | -6.21 | 12.4 | -6.79 | 5.9 | -7.80 | 160.3 | -5.69 | 862.3 | -5.63 |
| Youtube | 3.7 | -6.56 | 33.2 | -6.48 | 4.6 | -7.79 | 154.3 | -6.46 | 1814.3 | -6.42 |
| Pokec | 6.3 | -4.17 | 79.4 | -4.13 | 60.2 | -4.98 | 494.1 | -3.18 | 18656 | -3.17 |
| Orkut | 10.3 | -1.88 | 163.1 | -1.79 | 310.5 | -2.43 | 774.2 | -5.23 | - | - |
| Livejournal | 8.1 | -6.56 | 296.8 | -6.44 | 214.7 | -7.12 | 436.5 | -4.62 | - | - |
| Twitter | 33.7 | - | 625.0 | - | - | - | 2531 | - | - | - |

For OpMin, we compare our algorithm with the FAST forest sampling method [44]. To ensure a fair comparison, we set the number of forests in [44] to 1000, consistent with our experimental setup, and execute both algorithms in parallel using 8 threads. The EXACT algorithm, which uses the Laplacian solver, serves as the baseline method. Table 4 reports the running time and the average expressed opinion decline $\theta$ after setting the internal opinions of 50 nodes to zero, comparing three methods: PF-OPMIN, FAST, and EXACT. The results show that our proposed algorithm, PF-OPMIN, achieves comparable or even better effectiveness than FAST, while being significantly more efficient. For instance, on the LiveJournal network, PF-OPMIN is approximately 37 times faster than FAST, while also achieving a slightly larger opinion decline.

For the PDMin problem, we compare our proposed algorithm PF-PDMIN with the FASTGREEDY method [59], which utilizes the Johnson–Lindenstrauss Lemma [28] and a fast Laplacian solver [43]. In our experiments, we follow the parameter setting in [59], setting the size of the candidate edge set $E_C$ to $10^4$. And we use 20 iterations for the JL lemma. For our method, we fix $k = 50$ and $l = 1000$. Table 4 summarizes the running time and the reduction in polarization and disagreement, denoted by $\delta$, for both methods. The results demonstrate that PF-PDMIN is substantially more efficient and consistently achieves better performance than FASTGREEDY. For instance, on the Pokec network, our algorithm achieves more than a $35\times$ speedup while also yielding a greater decline in the polarization-disagreement index. Furthermore, FASTGREEDY fails to scale to the three largest networks due to time and memory constraints, whereas our method remains both efficient and scalable.

## 8 Conclusions

In this paper, we proposed several algorithms to compute and optimize opinion-based metrics for the Friedkin-Johnsen (FJ) model. We first introduced the concept of partial rooted forests and presented an efficient algorithm for computing several opinion-based quantities. Our proposed algorithms, PF-OPMIN and PF-PDMIN, reduce the time complexity from linear to sublinear compared to state-of-the-art methods. Extensive experiments on real-world networks demonstrate that our algorithms are both accurate and efficient, outperforming state-of-the-art methods and scaling effectively to large networks with over 20 million nodes.

Currently, our framework has limitations in handling optimization problems where the objective involves nonlinear terms such as squared components (e.g., $z_i^2$ in the marginal gain). In future work, we plan to extend our methods to support a broader class of opinion optimization problems under the FJ model.

## Acknowledgements

This work was supported by the National Natural Science Foundation of China (Nos. 62372112).

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

## A   APPENDIX

### A.1   Pseudocode of Algorithm PFS

---

**Algorithm 4:** PFS($\mathcal{G}, \boldsymbol{S}, l$)

---

**Input**   :Graph $\mathcal{G} = (V, E)$, node set $\boldsymbol{S} = \{v_1, \ldots, v_p\}$, number of samples $l$
**Output** :Partial rooted forest list $L$

1 $L \leftarrow [\,]$
2 **for** $k \leftarrow 1$ **to** $l$ **do**
3     $V_\phi \leftarrow \emptyset, E_\phi \leftarrow \emptyset$
4     **for** $i \leftarrow 1$ **to** $p$ **do**
5        $u \leftarrow v_i, P_u \leftarrow [\,], c \leftarrow u$
6        **while** *True* **do**
7           **if** $c \in V_\phi$ **then**
8              **break**
9           **else**
10              **if** *random()* $< \frac{1}{1+d_c}$ **then**
11                 $P_u \leftarrow P_u \cup \{c\}$
12                 mark $c$ as root node
13                 **break**
14              **else**
15                 $next \leftarrow$ random choice from $N_c^+$
16                 $P_u \leftarrow P_u \cup \{c, (c, next)\}$
17                 $c \leftarrow next$
18        $\hat{P}_u \leftarrow$ Apply loop-erasure to $P_u$
19        $V_\phi \leftarrow V_\phi \cup$ nodes in $\hat{P}_u$
20        $E_\phi \leftarrow E_\phi \cup$ edges in $\hat{P}_u$
21     Add $\phi = (V_\phi, E_\phi)$ to $L$
22 **return** $L$

---

### A.2   Proof of Lemma 5.1

**Proof.**   Since $z_i = \sum_{j=1}^n \omega_{ij} s_j$, the proof is reduced to showing $\mathbb{E}[\mathbb{I}_{\{r_\phi(i)=j\}}] = \omega_{ij}$. Consider the proof of Theorem 4.2, and we extend the partial rooted forest $\phi$ to a complete rooted forest $\phi'$. In $\phi$ and $\phi'$, we have $r_\phi(i) = r_{\phi'}(i)$ directly, since the nodes in $\phi$ are a subset of the nodes in $\phi'$. Using the Matrix Forest Theorem [12, 13], we have $\mathbb{E}[\mathbb{I}_{\{r_{\phi'}(i)=j\}}] = \omega_{ij}$. Thus, $\mathbb{E}[\mathbb{I}_{\{r_\phi(i)=j\}}] = \omega_{ij}$, and the estimator $\widehat{z}_i$ is unbiased.   $\square$

### A.3   Hoeffding's inequality

**Lemma A.1 (Hoeffding's inequality [25])** *Let* $x_1, x_2, \cdots, x_n$ *be* $l$ *independent random variables satisfying* $a \le x_i \le b$ *for all* $i = 1, 2, \cdots, n$. *Let* $x = \frac{1}{l} \sum_{i=1}^l x_i$. *Then for any* $\epsilon > 0$, $\mathbb{P}(|x - \mathbb{E}(x)| \ge \epsilon) \le 2 \exp\left(-\frac{2l\epsilon^2}{(b-a)^2}\right)$.

### A.4   Proof of Theorem 5.2

**Proof.**   Hoeffding's inequality, presented in Lemma A.1, serves as a powerful tool for estimating the required sample size $l$ to achieve a desired error guarantee. This inequality has been widely applied in various studies, such as [44, 39]. Utilizing Lemma 5.1 alongside Hoeffding's inequality, selecting $l = O(\frac{1}{\epsilon^2} \log \frac{1}{\delta})$ ensures that the estimated value $\widehat{z}_i$ deviates from the true value $z_i$ by at most an absolute error of $\epsilon$, with probability at least $1 - \delta$, for any node $i \in V$. Furthermore, according to Lemma 2.2 in Section 6 in [9], by choosing $p = O(\frac{1}{\epsilon^2} \log \frac{1}{\delta})$ and setting $\boldsymbol{S} = \{v_1, \cdots, v_p\}$,

we can guarantee that the sum $\sum_{i\in S} z_i$ approximates $\sum_{i\in V} z_i$ within an absolute error of $n\epsilon$, with probability at least $1 - \delta$. It is important to note that $z_i \in [0,1]$, by setting $l = O(\frac{1}{\epsilon^2} \log \frac{1}{\delta})$ and $p = O(\frac{1}{\epsilon^2} \log \frac{1}{\delta})$, the opinion-based measures $D(\mathcal{G})$, $P(\mathcal{G})$, $I(\mathcal{G})$, $C(\mathcal{G})$, and $DC(\mathcal{G})$ can be estimated within an absolute error of $n\epsilon$, with probability at least $1 - \delta$. $\square$

## A.5 Further Experiment Results

In this subsection, we report the standard deviations for two algorithms across five networks in Table 5 and provide further experimental results based on the exponential distribution in Table 6 and Table 7.

Table 5: Standard Deviation for two algorithms on five networks. Internal opinions are generated using the uniform distribution.

| Network | Standard Deviation in % | | | | | | | | | | | |
| | LazyWalk | | | | | | PF-QE | | | | | |
| | $z$ | $P$ | $I$ | $C$ | $DC$ | $D$ | $z$ | $P$ | $I$ | $C$ | $DC$ | $D$ |
|---|---|---|---|---|---|---|---|---|---|---|---|---|
| Delicious | 0.12 | 6.33 | 6.94 | 2.73 | 4.15 | 3.59 | 0.14 | 7.27 | 7.02 | 2.86 | 4.20 | 3.67 |
| Youtube | 0.006 | 0.93 | 0.70 | 0.13 | 0.27 | 2.18 | 0.009 | 0.98 | 0.73 | 0.17 | 0.30 | 2.04 |
| Pokec | 0.006 | 1.91 | 0.43 | 0.10 | 0.15 | 4.83 | 0.01 | 1.88 | 0.43 | 0.08 | 0.12 | 5.06 |
| Orkut | 0.02 | 4.02 | 0.91 | 0.16 | 0.71 | 44.83 | 0.02 | 3.32 | 0.92 | 0.25 | 0.47 | 24.17 |
| Livejournal | 0.006 | 1.15 | 0.55 | 0.14 | 0.33 | 4.84 | 0.05 | 1.17 | 0.59 | 0.43 | 0.36 | 4.60 |

Table 6: Mean relative errors and running times for three algorithms on six networks. Internal opinions are generated using the exponential distribution.

| Network | Time(s) | | | Reletive Error in % | | | | | | | | | | | |
| | LapSolver | LazyWalk | PF-QE | LazyWalk | | | | | | PF-QE | | | | | |
| | | | | $z$ | $P$ | $I$ | $C$ | $DC$ | $D$ | $z$ | $P$ | $I$ | $C$ | $DC$ | $D$ |
|---|---|---|---|---|---|---|---|---|---|---|---|---|---|---|---|
| Delicious | 6.6 | 2.3 | 2.1 | 2.50 | 2.79 | 3.09 | 1.66 | 1.17 | 3.81 | 2.41 | 1.96 | 3.05 | 0.79 | 1.10 | 3.75 |
| Youtube | 9.6 | 4.4 | 2.6 | 1.90 | 3.25 | 2.93 | 0.78 | 1.41 | 6.32 | 2.36 | 3.33 | 2.89 | 0.77 | 1.38 | 6.49 |
| Pokec | 48.5 | 42.7 | 5.4 | 3.35 | 3.72 | 2.89 | 0.48 | 1.32 | 10.91 | 2.50 | 3.64 | 2.72 | 0.43 | 1.30 | 10.85 |
| Orkut | 320.8 | 153.2 | 9.7 | 2.92 | 8.00 | 1.95 | 5.29 | 2.06 | 76.66 | 2.51 | 12.54 | 1.97 | 0.41 | 1.02 | 25.67 |
| Livejournal | 239.5 | 70.7 | 6.6 | 2.44 | 3.22 | 2.87 | 1.82 | 1.24 | 5.06 | 2.40 | 3.10 | 2.89 | 1.38 | 1.50 | 5.11 |
| Twitter | - | 168.6 | 34.9 | - | - | - | - | - | - | - | - | - | - | - |

Table 7: Standard Deviation for two algorithms on five networks. Internal opinions are generated using the exponential distribution.

| Network | Standard Deviation in % | | | | | | | | | | | |
| | LazyWalk | | | | | | PF-QE | | | | | |
| | $z$ | $P$ | $I$ | $C$ | $DC$ | $D$ | $z$ | $P$ | $I$ | $C$ | $DC$ | $D$ |
|---|---|---|---|---|---|---|---|---|---|---|---|---|
| Delicious | 0.008 | 1.49 | 2.15 | 0.59 | 0.87 | 3.07 | 0.01 | 1.37 | 2.12 | 0.57 | 0.89 | 3.00 |
| Youtube | 0.01 | 2.06 | 2.36 | 1.02 | 1.58 | 4.58 | 0.02 | 1.97 | 2.32 | 0.99 | 1.52 | 4.50 |
| Pokec | 0.01 | 3.29 | 1.95 | 0.33 | 0.87 | 8.12 | 0.02 | 3.24 | 1.87 | 0.34 | 0.883 | 7.63 |
| Orkut | 0.02 | 6.37 | 1.18 | 0.30 | 1.31 | 27.45 | 0.02 | 7.66 | 1.10 | 0.26 | 0.84 | 21.37 |
| Livejournal | 0.009 | 2.86 | 2.38 | 0.81 | 1.30 | 5.25 | 0.05 | 2.88 | 2.38 | 1.02 | 1.29 | 3.95 |

