# OpenReview forum: "Fast Computation and Optimization for Opinion-Based Quantities of Friedkin-Johnsen Model"
_NeurIPS.cc/2025/Conference — NeurIPS 2025 poster_

### Official Review · Reviewer_Q3ZZ · 2025-06-14

**Clarity:** 3
**Significance:** 2
**Originality:** 2
**Rating:** 4
**Confidence:** 4

**Summary:**

This paper proposes an efficient algorithm for computing the quantities in FJ model and applies the quantities estimation to the Opinion Minimization Problem and the Polarization and Disagreement Minimization Problem.

**Questions:**

Questions

1. In the experiments, for OpMin problem, the paper compares with FAST; for PDMin, the paper compares with FASTGREEDY. In the related work, the authors refer to many other recent baselines, which should be compared.

2. The experiments report Relative Error in the tables. In the greedy algorithms, the estimation error might lead to choosing different nodes or edges. Since the problems are usually not submodular, small estimation error could lead to completely different final results. It is better to present the final performance for OpMin and PDMin.

3. Some important references are missing, for example,

Sijing Tu and Stefan Neumann. A Viral Marketing-Based Model For Opinion Dynamics in Online Social Networks. WWW 2022.

Random-Walk Computation of Similarities between Nodes of a Graph with Application to Collaborative Recommendation, TKDE, 2007.

4. Small typos, e.g., “Reletive Error”->” Relative”

**Ethical Concerns:**

["NO or VERY MINOR ethics concerns only"]

**Final Justification:**

The authors have addressed my concerns. I slightly increase the score.

**Limitations:**

Please see the weakness of the questions.

**Quality:**

3

**Strengths And Weaknesses:**

Strengths:

1.	The motivation of partial rooted forests and the quantities estimation is clear.

2.	The paper provides a solid analysis of the performance guarantee.

Weaknesses:

1. The sampling based method is an extension of existing method. The existing method is sufficient to scale large graphs, which may limit the scope of readers.

2. The quantities estimation has been investigated previously and the method seems an application of the absorbing random walk, which limits the technical novelty.

---

> ### Author Rebuttal · Authors · 2025-07-30
>
> **Response to Reviewer Q3ZZ:**
>
> Thank you for your time and helpful comments. Below is a summary of our response to your concerns.
>
>
> **Response to Weakness 1:**
>
> We address three distinct problems related to the FJ model: (1) estimation of opinion-based quantities, (2) opinion minimization (OpMin), and (3) polarization and disagreement minimization (PDMin). Existing SOTA algorithms are tailored to each of these problems individually. In contrast, our work is the first to propose a unified framework capable of solving all three problems in sublinear time.
>
> Furthermore, for the most challenging case—PDMin, which is NP-hard—our method scales to very large graphs, including the Twitter dataset with over 40 million nodes. In contrast, the SOTA baseline FastGreedy fails to run on such large-scale data due to memory and runtime limitations. On other datasets, our method also demonstrates significant efficiency gains. For example, on Pokec (as shown in Table 3), our method achieves a speedup of approximately 40× compared to prior Laplacian solver-based methods, while maintaining comparable solution quality.
>
> **Response to Weakness 2:**
>
> In our paper, we introduce the concept of absorbing random walk (Section 4.1), which to our knowledge is the first formalization of this idea in the context of FJ models. It inspires the design of our partial rooted forest sampling method, where the complete rooted forest \[44] is a special case. When only a small subset of nodes is of interest, our method is significantly more efficient than either computing the full rooted forest or running absorbing random walks separately for each node, as shown in Theorem 4.2 (lines 510–514). Based on this, we develop a unified and scalable framework to solve three distinct FJ-related problems, each with different prior SOTA solutions.
>
>
> **Response to Question 1:**
>
> For FJ estimation, we compare our proposed algorithm with LapSolver \[52] and LazyWalk \[39]. For OpMin, we use the forest-based sampling method FAST \[44] as the baseline. For PDMin, we compare with FASTGREEDY \[59], a greedy algorithm based on a fast Laplacian solver. These baselines represent the current SOTA for the respective problems.
>
> Although many recent works have studied the FJ model, they do not directly address the three specific problems considered in our paper. Therefore, they are not suitable for comparison in our experimental setup. We hope for your understanding.
>
>
> **Response to Question 2:**
>
> In Table 2, for the estimation of opinion-based quantities, we report the relative error. For downstream tasks—OpMin and PDMin—we report the final performance in Table 3, measured by the decrease in opinion values and the P-D index, respectively. We will clarify this distinction more clearly in the revised version.
>
> **Response to Question 3 and Question 4:**
>
> We will add the two missing references in the revised version. We will also thoroughly check the entire paper and correct all typos. Thank you for your careful reading.
>
> Thank you again for your insightful suggestions.

---

> > ### Comment · Reviewer_Q3ZZ · 2025-08-03
> >
> > Thank you for your rebuttal. I will keep my score.

---

> > > ### Author Response · Authors · 2025-08-03
> > > **Response to Reviewer Q3ZZ**
> > >
> > > Thank you again for your time and help.
> > >
> > > We would like to kindly ask whether our rebuttal has addressed your concerns. If there are any remaining questions or suggestions, we would be grateful to hear them and would be happy to provide further clarification.

---

### Official Review · Reviewer_NDbc · 2025-07-02

**Clarity:** 2
**Significance:** 2
**Originality:** 1
**Rating:** 3
**Confidence:** 4

**Summary:**

This paper studies algorithms for efficiently computing the “final expressed opinions” in the standard Friedkin-Johnson (FJ) model of opinion dynamics. For a social network with Laplacian L, these opinions are equal to z = (L+I)^-1 s, where s is a vector of n “innate” opinions. The authors are specifically interested in computing some function of the opinions, like their inner product with a vector (with the all ones vector would give the average), their second moment z^Tz, and various other quantities related to “polarization and disagreement” of the final opinions. To do so, they adapt an approach based on random spanning forests from prior work. In particular, a natural approach to estimating a *single entry* is to write (L+I)^-1 as a power series in involving the random walk matrix D^-1A, exactly as is done when computing PageRank or personalized PageRank. The conclusion is that the i,j entry of  (L+I)^-1 is exactly equal to the chance that a random walk with absorption probability 1/1+d_k at node k that starts at i lands exactly on j. Thus, we can estimate entries in the ith row of (L+I)^-1 by running a bunch of random walks with this absorption probability, and seeing the percentage of times we land on various j. Recent work proposes and alternative approach based on the “Matrix Forest Theorem” which basically says that the probability an absorbing walk goes from j to i is also proportional to the probability i and j land in the same connected component if we construct a random spanning forest of G.

The authors observation is that, if we want to estimate a quantity like c^Tz for some vector v, we can subsample a set of indices, S, and return a Monte Carlo estimate. Accordingly, instead of building an *entire* random spanning forest for G, we should only build the part necessary to see what components nodes of S land in. This can be done easily by “early stopping” standard iterative algorithms for constructing random forests. The analysis of the result is thus straightforward (see Section A.3, A.4, A.5, A.6). Everything follows from prior work + Hoeffding’s inequality.

The authors show their methods can be used as subroutines to solve various optimization problems involving functions of the expressed opinions and compare the accuracy and runtime of their methods to prior work. There wasn’t sufficient discussion of the methods compared to — for example, it wasn’t clear to me what algorithm “LazyWalk” implements — however, it seems like the authors at least beat this baseline. At least in the main body also didn’t see to be a direct comparison to the “FAST” method (based on constructing a full random spanning forest), expect for on the downstream optimization problems. It would be helpful if the authors could comment on this, or provide results for FAST e.g. in Table 2.

**Questions:**

Mix of questions and small comments below. Don't feel the need to respond to everything:

- when discussing the prior work in Section 4, what is “l” when you say prior methods require O(ln) time?
- is “absorbing random walk” (e.g., as in Lemma 4.1) ever formally defined? At the beginning of Section 4.1 you define what it means for a walk to “absorb” at q, but not what it means to “be absorbing”.
- The transition from Section 4 to 5 could use work. At the end of 4, it’s not clear why Algorithm PFS is useful, or why we should care about it. In Theorem 4.2, what does it mean to be “more efficient than performing absorbing random walks for eahc node S”? At this point, the algorithm and the proposed alternative output different things — a partially rooted forest vs a bunch of walks. More efficient for what task or by what measure?
- Theorem 5.2 doesn’t seem scale invariant to me as stated. It implicitly depends on the assumption that s in [0,1], which was stated at some point, but not emphasized. I would remind the reader directly in the statement of Theorem 5.2.
- It would be helpful to clarify how Theorem 5.2 compares with what would be obtained with independent random walk samples. To me this would be a more meaningful comparison than what is currently in Theorem 4.2.
- It feels like to estimate something like z^Tz we should be able to randomly seleclt 1/epsilon^2 entries, and also get relative error to each entry with roughly 1/epsilon^2 walks. Average length is bar{d} so is the main imrpovement going from r to bar{d}?
- Could you add more details on the “average expressed minimm opinion problem”? In the *undirected setting* (I believe the settin of this work), I think it is well-known that the mean of z is equal to a weighted mean of s (entries weighted by degree). It seems like is should be straightforward to solve the problem — set the opinions to 0 for the k nodes with highest product of initial opinion and degree. The cited paper [44] handles the directed case, which is maybe more interesting? If the problem is only interesting in your setting when c is not the all ones vector, I think this is worth pointing out. It might also be helpful to clarify why vectors c are natural besides all ones.
- Some more discussion of what the baseline methods are in the experiments section would be helpful. E.g, what is te Lazy walk method?
- Does Lemma 5.1 require S to random? This isnt’ stated. Also, is it fine if Algorithm PFS is run on the elements of S in any order? I.e., the order doesn’t need to be random?

**Ethical Concerns:**

["NO or VERY MINOR ethics concerns only"]

**Final Justification:**

I raised my score by one because the authors addressed questions about comparisons with other methods, and how the $r$ dependence in their bounds compares with prior results depending on the average degree,  $\bar{d}$. Adding this discussion to the paper as well as the additional experiments will strengthen the work. However, I continue to be concerned about the relationship to [44] -- the authors methods is an easy modification of that work (add early stopping), which limits the novelty of this work.

**Limitations:**

yes

**Quality:**

1

**Strengths And Weaknesses:**

+ Interesting problem that has been of recent interest.
+ Reasonable algorithm that might very well be the "right" algorithm for the problem of estimating functions of the expressed opinions z.

- I don't see any technical lift of previous work -- unless I misunderstood, the main result amounts to an observation that you can early stop an algorithm proposed in prior work if you only care about estimating entries of z for a subset of nodes. This is a nice observation, but I think more would be required to elevate it to publication level.
- I found the writing informal and unclear at points. For example, the first theorem stated in the paper (Theorem 4.2)  is imprecise -- it is not clear in what way the partial random forest method can be directly compared to the efficiency of independent walk sampling, and the proof of the theorem doesn't make any such comparison.

---

> ### Author Rebuttal · Authors · 2025-07-30
>
> **Response to Reviewer sTM8:**
>
> Thank you for your time and helpful comments. Below is a summary of our response to your concerns.
>
>
> **Response to Weakness 1:**
>
> We focus on three problems related to the FJ model: (1) estimation of opinion-based quantities, (2) opinion minimization (OpMin), and (3) polarization and disagreement minimization (PDMin). SOTA algorithms for these three problems are different. For FJ estimation, we compare our proposed algorithm with LapSolver \[52] and LazyWalk \[39]. For OpMin, we compare with the forest sampling method FAST \[44]. For PDMin, we compare with the greedy algorithm FASTGREEDY \[59], which is based on a fast Laplacian solver. The time complexities of these algorithms are summarized in Table 1. Since these three problems have different properties, knowing a subset of entries of \$z\$ is not sufficient to solve them. Each problem requires a different SOTA method, and our work is the first to provide a unified framework to solve all these FJ-based problems in sublinear time.
>
> In our paper, we propose the concept of absorbing random walk (Section 4.1), which inspires the design of our partial rooted forest sampling method. The complete rooted forest \[44] is a special case of our method. In Theorem 5.2, we prove that it is not necessary to sample over all nodes to obtain accurate estimates of FJ opinion-based quantities. A similar idea of reducing sampling time appears in the empirical Bernstein bound \[\*].
>
> \[\*] Audibert, Jean-Yves, Munos, Rémi, and Szepesvári, Csaba. Tuning bandit algorithms in stochastic environments. International conference on algorithmic learning theory 2007.
>
> **Response to Weakness 2 and Question 3:**
>
> If we want to estimate one row of the forest matrix, absorbing random walk works well. If we want to estimate the entire forest matrix, then complete rooted forest is appropriate. However, if we only need the opinions of a small subset of nodes, our method is significantly more efficient than either running complete rooted forest or running absorbing random walk separately for each node. This is because our partial rooted forest reuses tree branches (lines 146-157), while repeated absorbing random walks discard prior computations and restart from scratch each time. We discuss this in the proof of Theorem 4.2 (lines 510–514), and we will provide more explanation in the revised version for clarity.
>
> **Response to Question 1 and Question 2:**
>
> In Section 4, \$l\$ denotes the number of samples used in forest-based sampling methods. We will make this explicit in the revised version. Regarding the term “absorbing random walk,” we use it to refer to a random walk that follows the transition rules specified in lines 110–114 until it is absorbed at a designated node. We will revise Section 4.1 to include a precise definition and improve clarity. Thank you for the careful observation.
>
> **Response to Question 4 and Question 5:**
>
> We mention in line 90 that $s \in [0,1]$ and will restate this in Theorem 5.2 to remind the reader. Thank you for the helpful suggestion. We propose the concept of absorbing random walk, which motivates the design of our partial rooted forest sampling method. In Theorem 4.2, we show that partial rooted forest sampling is more efficient than running absorbing random walks separately for each node. Therefore, Theorem 5.2 and the subsequent analysis focus on the partial rooted forest method.
>
> **Response to Question 6:**
>
> The transition from the absorbing random walk formulation to the partial rooted forest indeed improves the sampling complexity from $\bar{d}$ to $ r$. However, the key point is that the SOTA algorithms for the three FJ-related problems (estimation, OpMin, PDMin) are all different. For example, in the case of PDMin, prior work relies on Laplacian solvers, whereas our method achieves a speedup of approximately 40× in Pokec in Table 3 while obtaining comparable solution quality.
>
> **Response to Question 7:**
>
> In the undirected setting, the problem formulation in [44] becomes trivial since the mean of $z$ equals the degree-weighted mean of $s$. To address this, we introduce the weighting vector $c$, which generalizes the problem and makes it meaningful even in the undirected case. Importantly, our framework and the proposed partial rooted forest sampling method are applicable to directed graphs, and the estimator remains unbiased for the forest matrix in both directed and undirected settings. In this paper, we focus on undirected graphs because many FJ-related quantities, and the problem PDMin are defined and studied primarily in the undirected setting. Additionally, Laplacian solvers, which are used in the SOTA baselines for PDMin do not generalize to the directed case.  We will add more discussions on this point in the revised version.
>
> **Response to Question 8 and Question 9:**
>
> LazyWalk \[39] is the SOTA algorithm for estimating FJ opinion-based quantities. It performs multiple lazy random walks to estimate expressed opinions. We will add more discussions of baseline methods in the experimental section to make this clearer.
>
> Regarding Lemma 5.1, the subset $S$ is assumed to be randomly selected, and we will make this assumption explicit in the revised version. As for Algorithm PFS, the order of nodes in $S$ does not affect the result; it can be executed in any order.
>
> **Additional Response:**
>
> The three FJ-related problems we address—estimation, OpMin, and PDMin—each have different SOTA baselines. Among them, only OpMin uses a forest-based method (FAST) as the baseline. Nonetheless, for completeness, we also applied the full rooted forest construction to the estimation task and report its runtime in Table 2 below:
>
> | Network     | Algorithm           | Time  | P      | I      | C      | DC     | D      |
> |-------------|---------------------|-------|--------|--------|--------|--------|--------|
> | Delicious   | PF-QE | 2.2   | 1.38   | 0.42   | 0.55   | 0.32   | 4.05   |
> |             | Full Forest         | 4.1   | 4.57   | 2.01   | 0.28   | 0.68   | 1.96   |
> | Youtube     | PF-QE               | 2.4   | 0.95   | 0.73   | 0.22   | 0.31   | 4.19   |
> |             | Full Forest         | 8.5   | 4.14   | 1.93   | 0.48   | 0.11   | 5.84   |
> | Pokec       | PF-QE               | 5.3   | 2.01   | 0.96   | 0.21   | 0.51   | 10.63  |
> |             | Full Forest         | 26.4  | 3.12   | 1.23   | 0.73   | 0.20   | 16.3   |
> | Orkut       | PF-QE               | 9.9   | 2.98   | 0.72   | 0.23   | 0.47   | 13.67  |
> |             | Full Forest         | 66.8  | 2.81   | 1.06   | 0.75   | 0.20   | 18.8   |
> | Livejournal | PF-QE               | 6.9   | 1.30   | 0.90   | 0.30   | 0.52   | 8.17   |
> |             | Full Forest         | 58.2  | 3.51   | 1.45   | 1.41   | 0.55   | 15.9   |
>
> We sample 100 full forests and record the runtime and relative error. As shown in the table, our partial forest (PF) method is consistently faster than the full forest (FF) method, while also achieving better estimation accuracy.
>
> Thank you again for your insightful suggestions.

---

> ### Comment · Reviewer_NDbc · 2025-08-04
>
> Thanks for the responses and the additional experimental results. For Theorem 5.2, it might make sense to add to the paper a more direct comparison of how the result compares to what is achievable with a random walk based method, and how we expect r and bar{d} to compare.
>
> Edit: Sorry, I see that you added a comment comparing r and bar{d} to another reviewer. I think the table provided would be great to add to the paper, and would give the reader a more concrete understanding of the possibly benefit over random walk based methods.

---

> > ### Author Response · Authors · 2025-08-05
> > **Response to Reviewer NDbc**
> >
> > Thank you for your response and valuable suggestions. We will include the additional experimental results, along with the comparison table of $r$ and $\bar{d}$, in the revised version of our paper.
> >
> > If you have any further questions or suggestions, we would be happy to provide further clarification.

---

### Official Review · Reviewer_tebe · 2025-07-02

**Clarity:** 4
**Significance:** 3
**Originality:** 4
**Rating:** 5
**Confidence:** 5

**Summary:**

The paper provides a cost-efficient way to compute and optimize opinion-based quantities in the Friedkin–Johnsen (FJ) social-influence model. Central to these tasks is the forest matrix $\Omega$, which has connections to absorbing random walks. The authors present a method for approximating the final opinions $z$ in FJ model by providing a sampling method. Building on Wilson's loop‑erased random‑walk sampler, they introduce Partial Forest Sampling (PFS), a local algorithm that generates only the few forest components needed for a given query. Unbiasedness and Hoeffding-type error bounds guarantee that averaging over all sampled forests yields accurate estimates of node opinions, polarization, and disagreement.

Leveraging the same sampler, the paper presents two optimization algorithms: OPMIN, which selects $k$ internal opinions to minimize overall deviation from truth, and PDMIN, which adds up to $k$ edges to minimize polarization and disagreement index, which is convex in nature. Both achieve sub-linear expected runtimes and come with provable approximation guarantees, and it's a good asymptotic improvement over earlier approaches.

**Questions:**

Can you provide some insights if this can also be employed in approximately measuring polarization index which is non-convex in nature?

**Ethical Concerns:**

["NO or VERY MINOR ethics concerns only"]

**Final Justification:**

I am keeping my score. While the proposed algorithms are built on an interesting observation, their efficiency and scalability remain noteworthy.

**Limitations:**

yes

**Paper Formatting Concerns:**

No Major paper formatting concerns

**Quality:**

4

**Strengths And Weaknesses:**

Strengths:

1) Sublinear time algorithms

2)  Practical scalability across various datasets

Overall the paper is a well‑written contribution that combines sound theory with convincing large‑scale experiments.

Weakness:

1) Maybe I missed it but the choice of $r$ is not very clear from the text. Can the authors elaborate on it?

---

> ### Author Rebuttal · Authors · 2025-07-30
>
> **Response to Reviewer tebe:**
>
> Thank you for your time and helpful comments. Below is a summary of our response to your concerns.
>
>
> **Response to Weakness:**
>
> The parameter $r$ is determined by the structure of the graph. As shown in Lemma 4.1, its maximum value is bounded by the average degree $\bar{d}$. In practice, for real-world social networks (as reported in Table 4), we observe that $r < 15$ across all datasets. Furthermore, we conducted additional experiments on structured, regular, and scale-free graphs, consistently finding that $r$ remains small in all cases.
>
> | Graph Type            | Nodes  | Avg Degree ($\bar{d}$) | Observed $r$ |
> | --------------------- | ------- | ---------------------- | ------------ |
> | Grid (500×500)        | 250,000 | 3.992                  | 3.49         |
> | 15-Regular Graph      | 100,000 | 15.000                 | 3.69         |
> | Barabási–Albert Graph | 100,000 | 19.998                 | 3.04         |
>
> **Response to Question:**
>
> As shown in Table 2, our method provides accurate estimates of the polarization index $P$ in sublinear time, with relative error below 3%. Therefore, if an optimization problem is formulated based on polarization or a related non-convex index, our framework can still be used to estimate the objective efficiently. Extending the optimization aspect to non-convex formulations is a promising direction for future work.

---

> > ### Comment · Reviewer_tebe · 2025-08-02
> >
> > Thank you for your rebuttal. I will keep my score.

---

### Official Review · Reviewer_poCz · 2025-07-09

**Clarity:** 4
**Significance:** 4
**Originality:** 4
**Rating:** 5
**Confidence:** 3

**Summary:**

This paper proposes Partial Rooted Forest Sampling (PFS), a fast sublinear-time method for estimating key quantities in the Friedkin-Johnsen opinion dynamics model. By using loop-erased absorbing random walks to construct partial rooted forests, the method enables efficient, unbiased approximation of the forest matrix and related opinion metrics. It also supports fast optimization for tasks like reducing overall opinion or polarization, outperforming prior methods on large-scale real-world networks.

**Questions:**

Your algorithm’s performance depends critically on the graph-dependent ratio r, which is shown empirically to be small. Could you elaborate on which structural properties of graphs influence r, and how it behaves on different types like expanders or grid graphs?

Algorithm 1 samples S randomly, while Algorithm 3 uses endpoints of candidate edges, suggesting that smarter sampling could help. Could a more strategic selection of S, such as high-degree or central nodes, reduce the number of samples p or forests l needed for accurate estimation?

**Ethical Concerns:**

["NO or VERY MINOR ethics concerns only"]

**Limitations:**

yes

**Quality:**

4

**Strengths And Weaknesses:**

Strengths:
The PFS method is a novel and elegant idea that localizes sampling to relevant graph regions, reducing complexity from linear to sublinear and enabling practical analysis of massive networks. The paper is theoretically solid with clear proofs, and the experiments show dramatic speedups and superior scalability compared to strong baselines. Besides, the paper is exceptionally well-written, with clear explanations, consistent notation, and well-designed visuals that effectively support its claims.


Weaknesses:
The method’s efficiency relies on a parameter r that depends on the graph structure. Although empirical results show r is small in practice, a deeper theoretical analysis of how graph properties affect r would enhance understanding and applicability.
The paper briefly mentions limitations only in the conclusion. Including a dedicated "Limitations" section with discussion on issues like variance in Disagreement estimation and nonlinear objectives would improve transparency and completeness.

---

> ### Author Rebuttal · Authors · 2025-07-30
>
> **Response to Reviewer poCz:**
>
> Thank you for your time and helpful comments. Below is a summary of our response to your concerns.
>
>
> **Response to Weakness and Question 1:**
>
> As shown in Lemma 4.1, the parameter $r$ is upper bounded by the average degree $\bar{d}$, but in practice we find that $r$ is often much smaller, depending on the structural properties of the graph. To better understand what influences $r$, we evaluated it on several graph families with diverse characteristics. The table below reports the observed values:
>
> | Graph Type            | Nodes  | Avg Degree ($\bar{d}$) | Observed $r$ |
> | --------------------- | ------- | ---------------------- | ------------ |
> | Grid (500×500)        | 250,000 | 3.992                  | 3.49         |
> | 15-Regular Graph      | 100,000 | 15.000                 | 3.69         |
> | Barabási–Albert Graph | 100,000 | 19.998                 | 3.04         |
>
> These results show that $r$ remains small across a variety of graph types, including structured, regular, and scale-free graphs. We will elaborate further on this point in the revised version. Thank you again for the helpful suggestion.
>
> **Response to Question 2:**
>
> In Algorithm 1, the subset $S$ is sampled randomly because the proof of Theorem 5.2 relies on the randomness of $S$ to guarantee unbiasedness and concentration. In contrast, in Algorithm 3, we set $S$ to be the endpoints of candidate edges, since we are only interested in the opinions of those specific nodes.
>
> Thank you for the insightful suggestion. We will include a discussion on how a more strategic selection of $S$, such as using high-degree or central nodes, might affect the performance of downstream tasks like PDMin in our revised version.

---

### Note · Authors · 2025-08-12

We sincerely thank all reviewers, the area chairs, and program chairs for their time, effort, and constructive feedback on our submission.

Our paper addresses the fast computation and optimization of opinion-based quantities in the Friedkin–Johnsen (FJ) model. We introduce the concept of partial rooted forests and develop a unified sampling-based framework that solves three important problems—(1) estimation of opinion-based quantities, (2) opinion minimization (OpMin), and (3) polarization and disagreement minimization (PDMin)—all in sublinear time. Our methods capture essential structural information while scaling to very large graphs (e.g., the Twitter network with over 40 million nodes), achieving significant speedups over existing state-of-the-art approaches while maintaining competitive accuracy.

The main concerns raised by the reviewers focus on two points: (1) providing more experimental evidence for the parameter $r$, and (2) clarifying the relationship between our algorithms and the baselines/SOTA methods. In response to the first point, we supplemented our experiments with additional evaluations on multiple graph families and real-world networks. As shown in Lemma 4.1, $r$ is upper bounded by the average degree $\bar{d}$, but is typically much smaller in practice. Our new experiments confirm that $r$ remains small across diverse settings—for example: 15-Regular Graph: $\bar{d} = 15.000$, $r = 3.69$; Barabási–Albert Graph: $\bar{d} = 19.998$, $r = 3.04$. Similar trends hold for all tested real-world networks ($r < 15$).

For the second point, we clarified that our work addresses three distinct problems under a unified framework, whereas prior SOTA methods are tailored to only one of these problems individually. In particular, for the most challenging PDMin problem—which is NP-hard—our method scales to massive networks where the SOTA baseline FastGreedy fails due to memory/runtime limits. On medium-scale networks, our algorithms achieve substantial efficiency gains (e.g., a 40× speedup on Pokec for OpMin) while maintaining similar or better solution quality.

In the rebuttal stage, no reviewer raised new questions. For the final version, we will (i) elaborate further on the behavior of $r$ and its practical implications, (ii) provide a clearer explanation of our algorithms relative to baselines, (iii) include additional experimental results and visualizations.

Once again, we thank the reviewers and chairs for their valuable feedback.

---

### Decision · Program_Chairs · 2025-09-17

**Decision:**

Accept (poster)

**Comment:**

This work studies the problem of efficient estimation of opinion-based quantities as well as optimization of opinion-based objectives (minimizing the weighted average final expressed opinion and reducing disagreement between nodes) on social networks under the well-studied Friedkin-Johnsen model. The authors extend existing algorithms using what they refer to as a partial rooted forest sampling approach based on results from [44, 45, 46] in their references. Effectively, their method stops the method from [44] early, thereby avoiding full graph traversal and reducing the time complexity from linear to sublinear.

The reviewers agree that the main contributions of this work provide an interesting and elegant observation about existing methods discussed in the papers above. However, the reviewers also expressed numerous concerns about technical novelty since the presented algorithms are a natural extension of the existing ones. Of course, extensions can make a meaningful and interesting contribution, but such papers benefit from providing substantially more empirical insights across a broader family of real-world networks, more robustness results (e.g., around the parameter r which is crucial to their methods performance), a more extensive direct comparison with existing methods and natural baselines, and other technical extensions of the problem such as non-linear objectives as the authors mention. The additional results presented in the rebuttal phase helped mitigate some of these concerns, but some concerns remain. We encourage the authors to incorporate to detailed feedback from the reviewers.